# Socrates and the Sophists: Reconsidering the History of Criticisms of the Sophists

Noburu Notomi

Department of Philosophy, Graduate School of Humanities and Sociology, University of Tokyo, Tokyo 113-8654, Japan; notomi@l.u-tokyo.ac.jp

**Abstract:** To examine the sophists and their legacy, it is necessary to reconsider the relation between Socrates and the sophists. The trial of Socrates in 399 BCE seems to have changed people's attitudes towards and conceptions of the sophists drastically, because Socrates was the first and only "sophist" executed for being a sophist. In the fifth century BCE, people treated natural philosophy, sophistic rhetoric and Socratic dialogue without clear distinctions, often viewing them as dangerous, impious and damaging to society. After the trial of Socrates, however, Plato sharply dissociated Socrates from the sophists and treated his teacher as a model philosopher and the latter as fakes, despite many common features and shared interests between them. While Plato's distinction was gradually accepted by his contemporaries and by subsequent thinkers through the fourth century BCE, some disciples of Socrates and the second generation of sophists continued to pride themselves on being sophists and philosophers at the same time. Thus, this paper argues that Socrates belonged to the sophistic movement before Plato dissociated him from the other sophists, although the trial of Socrates did not immediately eliminate confusion between the sophist and the philosopher. The reconstructed view of the contemporaries of Socrates and Plato will change our conception of the sophists, as well as of Socrates. Finally, the paper examines the relation of Socrates to Antiphon of Rhamnus. Plato deliberately ignored this Athenian sophist because he was a shadowy double of Socrates in democratic Athens.

**Keywords:** sophists; Socrates; Plato

## 1. Discussing Socrates the Sophist

Sophists made up the cultural climate in Athens from the mid-fifth century BCE to the fourth century BCE, but it was not until 399 BCE that the sophists became a philosophical problem. Early in the spring of that year, Socrates, an Athenian citizen and well-known intellectual, was summoned to court for impiety and sentenced to death as a dangerous intellectual called a "sophist". Today, however, we believe that Socrates was truly a philosopher and by no means a sophist. I call this modern belief "the traditional view of the history". In this essay, I will reconsider the relation between Socrates and the sophists to clarify the issue of whether sophists were false philosophers to be criticized. My main point will be that Socrates belonged to the sophistic movement before Plato dissociated him from the other sophists. The reconstructed viewpoint of the contemporaries of Socrates and Plato will change our conception of the sophists as well as of Socrates.

Let us first examine the traditional view of the history in two respects. First, the intellectuals, called sophists, were active in Greece even before the trial of Socrates. Starting with Protagoras of Abdera (one generation senior to Socrates), such thinkers as Prodicus of Ceos, Hippias of Elis, Thrasymachus of Chalcedon and Gorgias of Leontini were usually numbered among the sophists.[1] Although Protagoras's self-introduction as "the first sophist" in Plato's eponymous dialogue may be doubted for its historical accuracy, with some believing it a fictional account by the author (Plato, *Protagoras* 317B [DK 80 A5]; *Suda* Π 2958; cf. Notomi 2013, pp. 12–14), we can be sure that people of the late fifth century

BCE used the word "sophists" primarily to refer to the intellectuals Plato depicted in his dialogues. Second, the trial of Socrates in 399 BCE seems to have changed people's attitudes towards and conceptions of the sophists drastically, because Socrates was the first and only "sophist" executed for being a sophist—i.e., one who did not believe in traditional gods and who corrupted the young. After the trial, people had to face two questions: was Socrates a sophist, and should sophists be executed? I will show that the trial of Socrates did not immediately eliminate confusion between the sophist and the philosopher, and that since the trial the term "sophist" has become truly controversial, whether used positively, as by Isocrates and Alcidamas, or negatively, as by Plato and Aristotle in the fourth century BCE. An important question is how the name and notion of "sophist" became problematic.

Therefore, this essay asks two questions that are usually contemplated separately: What is a sophist, and who is Socrates? The two questions are deeply interconnected, or the answers to these questions may be two sides of one coin, whereas they are usually discussed separately. On the one hand, scholars of Socratic and Platonic studies have strongly resisted associating Socrates with the sophists. They separate and compare them, where one is positive and the other negative. Historians of rhetoric, on the other hand, have hesitated to place Socrates among the sophists, since he was the harshest critic of rhetoric in Plato's dialogues. Both tend to exclude one of the two questions from their contemplations for different reasons. Thus, I think we should combine the two questions to provide a single answer, allowing an examination of the traditional view.

A paradigmatic treatment of the two questions is seen in W. K. C. Guthrie's *A History of Greek Philosophy*, published in six volumes from 1962–1981. While Guthrie originally discussed both in the third volume titled *The Fifth-Century Enlightenment* in 1969 (Guthrie 1969), he soon reprinted it in two separate paperback volumes in 1971. Part One, "The World of the Sophists", became *The Sophists* (Guthrie 1971a), and Part Two, "Socrates", became *Socrates* (Guthrie 1971b).[2] The two parts in the original edition were generally independent, so we see that the second part scarcely mentions the sophists, while the first part includes scattered references to Socrates. It seems that the two parts were combined mainly for chronological reasons.

Next, George Kerferd, in his monograph concerning the sophists, *The Sophistic Movement*, argues that Socrates should be included as part of this movement (Kerferd 1981a, pp. 55–57). In his analysis of the idea of "Socrates as a sophist" in the chapter "The Individual Sophists", he claims that "Socrates was quite widely regarded as part of the sophistic movement" and that "the fact that he took no payment does not alter his function in any way" (Kerferd 1981a, p. 57). By challenging the general assessment of the historical sophists, Kerferd insisted that they were serious thinkers and played important roles in philosophy by engaging in a wide range of scientific fields. For example, he describes the sophists as "pioneer thinkers" in sociology and other fields (Kerferd 1981a, p. 57). In this respect, Kerferd saw no reason why Socrates should be excluded from this group of philosophers. Kerferd discusses this idea again in the article included in his edition of *The Sophists and their Legacy* (Kerferd 1981b, 1981c). Here he emphasizes that "among them [the sophists] also was Socrates" (Kerferd 1981b, p.4), because the problems discussed and methods used were shared by Socrates and the sophists. By treating Socrates and the sophists as related, however, he eliminated the opposition or contrast between them. In this sense, Kerferd suggested that the sophists, including Socrates, were philosophers, as opposed to the view that Socrates was a sophist in the negative sense.

Kerferd's suggestion was not taken seriously for a few decades, and scholars never abandoned or reconsidered the separation and contrast between Socrates and sophists. However, it was eventually introduced in the new source books of André Laks and Glenn Most's *Early Greek Philosophy* (Laks and Most 2016), published in nine volumes, among which the eighth volume, "Sophists Part 1", discusses Socrates in the third chapter, thus changing his location in the history of philosophy.[3] Laks and Most include eleven thinkers in the two volumes of sophistic texts: the eighth volume includes chapters devoted to Protagoras, Gorgias, Socrates, Prodicus, Thrasymachus and Hippias, while the ninth

volume includes chapters on Antiphon, Lycophron, Xeniades, as well as the *Anonymous of Iamblichus*, *Pairs of Arguments* (*Dissoi Logoi*), and a chapter titled "Sophists and Sophistic: Collective Representations and General Characterisations".

Although the intention of the editors is not entirely clear, it accords with Kerferd's suggestion to include Socrates among the sophists.[4] Their decision to classify Socrates as a sophist may have a tremendous impact on the traditional view of Greek philosophy in two aspects. First, Socrates used to be regarded as the starting point of authentic philosophy, and from this view, the preceding thinkers, namely, natural philosophers *and* sophists, were called "PreSocratics". Although it has repeatedly been noted by historians that this designation is inappropriate, both in the chronological sense and the philosophical treatment, it is still customary to accept this schema.[5] *Die Fragmente der Vorsokratiker* (DK), edited by Hermann Diels and Walther Kran in 1903/1952, placed the section *Ältere Sophistik* after the natural philosophers in the second volume (Diels and Kranz 1952). It includes twelve chapters: "*Name und Begriff*", "*Protagoras*", "*Xeniades*", "*Gorgias*", "*Lykophron*", "*Prodikos*", "*Thrasymachos*", "*Hippias*", "*Antiphon der Sophist*", "*Kritias*", "*Anonymus Iamblichi*" and "Δισσοὶ λόγοι (*Dialexeis*)". By contrast, Laks and Most improved this earlier treatment by placing Socrates among the sophists, together with some changes to the order and the omission of a few minor chapters.[6] Their chapter on Socrates collects relevant testimonies concerning the life and doctrines of Socrates (though no fragments of his writings, of course), which were not included in the first Diels and Kranz edition. However, the testimonies concerning Socrates and his circle were collected extensively in Gabriele Giannantoni's *Socratis et Socraticorum Reliquiae* (*SSR*), published in four volumes in 1990. Again, we should note that *Socratis et Socraticorum Reliquiae* does not discuss the sophists as such, while the Laks and Most edition does not include much information about the pupils of Socrates.[7]

These treatments of the sophists and Socrates by major historians indicate a fundamental difference in how to view the history of Greek philosophy. They raise two crucial questions: whether we treat Socrates as part of the sophistic movement and whether we include the sophists among the PreSocratics. On the one hand, just as the new treatment by Laks and Most implies, it will make a crucial difference whether Socrates is discussed together with the sophists in treatments of early Greek philosophy. On the other hand, I suggest that a major division should be made between the natural philosophers in Ionia and Italy and the sophists and Socrates, active in Athens, where the former should be called "Early Greek Philosophy" (in the sixth and fifth centuries BC) and the latter "Classical Greek Philosophy" (from the mid-fifth century BCE to the late fourth century BCE).[8] For a more careful treatment of these thinkers, we must examine the first two stages of Socratic testimonies, before and after his death in 399 BCE. It will become clear that views were radically different in these two stages.

## 2. Before and After Socrates

The question of who Socrates was has long been central to the history of philosophy. Socrates is deemed to be one of the greatest philosophers, or even the most important philosopher, who influenced or even founded the field of philosophy. Scholars tend to see something special in this thinker. In particular, Socrates is viewed as representing a turning point in the history of ancient Greek philosophy, if not the birth of genuine philosophy. The following famous statement by Cicero has been cited repeatedly by historians of philosophy: "But Socrates was the first one who brought down philosophy from the heavens, established it in cities, introduced it into families, and forced it to examine life and morals, and good and evil" (Cicero, *Tusculan Disputations* 5.4.10 = LM SOC. D4 [trans. Laks and Most]). Cicero's statement represents the typical view that separates Socrates from his predecessors in the field of interest, and his authority helped generate the common view that philosophers before Socrates discussed only natural phenomena and should thus be regarded as natural scientists more than philosophers. Cicero's comment also contributed to the creation of the image of Socrates as an innovator: he was the first thinker who examined ethics and should therefore be called a philosopher. However, Cicero did

not create this image, but rather adopted earlier views of Socrates. Cicero's report returns, for example, to the testimony of Xenophon's *Memorabilia*, which defends Socrates's piety against the accusation of Polycrates.[9] Xenophon explains the reason as follows: "For unlike most of the others he did not discuss the nature of the universe or consider how what the sophists call 'the *kosmos*' is constituted, or by what governing laws the heavenly phenomena come about, but he would prove that those who ponder such matters are fools" (Xenophon, *Memorabilia* 1.1.11; trans. M. D. Macleod, modified).

Xenophon distinguishes Socrates from those who discussed "nature [*physis*]", meaning the natural philosophers in Ionia and Italy, and in this sense, his views accord with Cicero's contrast between "heavens" and "cities". However, the expression "what the sophists call 'the *kosmos*'" is problematic. The term *kosmos* was used in the sense of the "ordered world" or "universe", originally by Pythagoras and the Pythagoreans, but it was not a major topic among the sophists when using the term to refer to professional teachers. Protagoras, Hippias and others did indeed discuss some issues of natural philosophy and mathematics, but such topics were far from their main concern. However, Xenophon knew the standard meaning of "sophist", so it seems unlikely that he used the word in the general sense of "professor" or "philosopher" or confused it with natural thinkers, although most commentators view it this way.[10] Moreover, the verb "ponder [*phrontizein*]" used in this text reminds us of the sophistic pondering at Socrates's school, called the *phrontistērion* or "thinking-shop", in Aristophanes's *Clouds* (94). Therefore, I suspect that Xenophon's reference to the sophists here reflects the original charge against Socrates as the old accusation in Plato's *Apology of Socrates*: "There is a certain Socrates, a wise [*sophos*] man, a ponderer [*phrontistēs*] over the things in the air and one who has investigated the things beneath the earth and who makes the weaker argument the stronger [*ton hēttē logon kreittō*]" (Plato, *Apology of Socrates* 18B; trans. H. N. Fowler; a similar sentence is presented in 19B).

This accusation is ambiguous and appears to be a sheer confusion of different elements, especially between natural philosophers and sophists. Celestial phenomena are a typical subject of natural philosophers. Anaxagoras was a representative natural philosopher who is also mentioned in the *Apology* (26D). He was sometimes called a sophist who was guilty of impiety (Diodorus Siculus, *Historical Library* 12.39.2 = DK 59 A17 = LM ANAXAG. P24), and he is said to have been accused of impiety for claiming that the sun was a burning stone (Diogenes Laertius, *Lives of the Eminent Philosophers* 2.12–14 = DK 59 A1 = LM ANAXAG. P23). While there is little evidence of investigation into things beneath the earth among natural philosophers, it must have been a popular opinion that natural philosophy discussed what exists both above and below our living region, which people believed were the spheres of the gods and the dead, respectively. Thus, the charge of impiety was strongly associated with pondering about natural phenomena, and both the natural philosopher Anaxagoras and the sophist Protagoras are said to have been accused of impiety in Athens (Plutarch, *Nicias* 23 = DK 59 A18 = LM ANAXAG. P25b; Libanius, *Defence of Socrates* 154).

Moreover, the phrase "make the weaker argument the stronger" (*ton hēttō logon kreittō poiein*) reflects the sophists' skill in speech, represented by Protagoras and criticized by Aristotle in the *Rhetoric* under the name "Protagoras's claim".[11] In Aristophanes's *Clouds*, Socrates teaches in his school "both stronger and weaker arguments" (*kreittōn kai hēttōn logoi*), and the protagonist of the play, Strepsiades, notes that Socrates claims that he can "plead the unjust side of a case and win" (112–115; DK 80 C 2). Later in the play, the personified Weaker Argument (*ho hēttōn logos*) actually defeats the Stronger Argument (*ho kreittōn logos*) (889–1111). Thus, the phrase reflecting Protagoras's skill in argumentation is associated with Socrates in the *Clouds* and mentioned in the old accusation in Plato's *Apology*. Aristophanes is the only name that Socrates mentions as an old accuser in the *Apology* (18C–D, 19C). The first version of the *Clouds* was produced in 423 BCE, and because it received the third and last prize, the poet rewrote the play, and the revised version has been transmitted to the present day. In the extant works, Aristophanes rarely uses the word "sophist", but in three places in the *Clouds* the term clearly refers to the teaching activity of

Socrates;[12] in particular, Socrates and Prodicus are mentioned together as "meteorosophists" (360), while a similar term, "*meteorologoi*", appears in the context of sophistical arguments in Gorgias's *Encomium of Helen* (13). Although we know little about the personal relationship between Aristophanes and Socrates,[13] the comic play featuring Socrates reinforced the popular image of Socrates as a natural philosopher and professional sophist.

While modern readers tend to see two distinct kinds of intellectual activities in the old accusation of Plato's *Apology*, as well as in Aristophanes's *Clouds*, we should understand that ordinary people's conception of the wise man at the time of Socrates's trial (399 BCE) included both elements—natural philosopher and sophist—without a clear distinction. Both are pursued by wise men who know many subjects and have excellent skill in speaking. The charge of impiety was directed to both: natural philosophers investigated natural phenomena, which people believed to be divine entities, and sophists cast doubts on the existence of the gods, as Protagoras and Prodicus did. Both kinds were regarded as dangerous to young citizens in that they denied traditional religious beliefs.

The old accusation in Plato's *Apology* has much in common with the explanation of Xenophon, since both authors adopt a similar strategy in defending their master. In Plato, Socrates clarifies, against the old accusation, that he was neither a natural thinker nor a professional teacher. First, Socrates insists that he had never discussed such issues and asks the jurors to be witnesses to each other (*Apology* 19 C–D). And second, after quickly dismissing this first charge, he advances a longer argument to show that he was not a sophist who took fees to teach young people, as did those like Gorgias, Prodicus, Hippias and Euenus of Paros (19D–20C). Socrates thus provides separate arguments to deny that he possessed the wisdom of these two kinds of wise men.

In the *Memorabilia*, Xenophon similarly argues in two stages. He first states that Socrates always discussed human issues, such as "what is beautiful" and "what is justice" (1.1.16). After he dissociates Socrates from "those who worry with 'nature of the universe [*hē tōn pantōn physis*]'" (1.1.14), he adds that Socrates encouraged his companions to care for the soul and that he did not "make money himself out of their desire for his companionship". Without using the word "sophist", Xenophon here separates Socrates from the sophists, who charged fees for teaching: "He [Socrates] held that this self-denying ordinance insured his liberty. Those who charged a fee for their society he denounced for selling themselves into bondage; since they were bound to converse with all from whom they took the fee" (1.2.6). Later in the text Xenophon associates this defence with a conversation between Socrates and the sophist Antiphon (1.6). Thus, Xenophon connects the charge of corrupting the youth with the image of Socrates as a sophist, which must have been the focus of his trial.

Despite the excuses of his pupils, however, Socrates continued to be criticized as a dangerous sophist in Athens. The Athenian sophist Polycrates seems to have called Socrates a sophist in his critical pamphlet around 393 BCE,[14] and a half century later, around 346–345 BCE, the rhetorician Aeschines called Socrates a sophist in his forensic speech, *Against Timarchus*: "Did you put to death Socrates the sophist, fellow citizens, because he was shown to have been the teacher of Critias, one of the Thirty who put down the democracy?" (173; trans. C. D. Adams). This reference to Socrates is intended to highlight the danger and vice of Demosthenes in corrupting the young and harming Athenian democracy. Aeschines later calls Demosthenes a "sophist" (175). Although his opinion of Socrates is unclear, Aeschines uses this accusation as if it was a common opinion about Socrates.

This survey has shown that the traditional view of Socrates we have today differs radically from what people thought of him in antiquity. We should therefore carefully distinguish between the historical position of Socrates in his own time and in later periods. Testimonies about Socrates may be grouped into three categories. First, there are several contemporary references to Socrates in the fifth century BCE, when he was still alive. Most are in comedies, one of which is by Aristophanes, who ridiculed this strange thinker.[15] In this first stage, Socrates was included among the sophists. Second, numerous works were

written after his death that featured Socrates or were about Socrates. Most are "Socratic dialogues" written by his pupils, including Plato and Xenophon, to defend him. We also know that Socrates was harshly criticized, especially by Polycrates in his *Accusation of Socrates*, whose critical arguments can be partly reconstructed from Xenophon and Libanius. In the second stage, then, works on Socrates were written by those who knew him. The writers of the third stage (after the mid-fourth century BCE), however, spoke about Socrates without having been acquainted with him. For example, Aristotle, born 15 years after Socrates's death, comments on his contributions to philosophy in several of his works, and Aristoxenus reproaches him in his *Life of Socrates*. Much later, writers, such as Epictetus in the second century CE, in admiration of Socrates's life, spoke of him based on the Socratic literature.

We should distinguish among these three stages and use the testimonies with an understanding of each group in terms of both values and bias. Accordingly, the views on sophists should also be distinguished in these stages. Thus, in the fifth century BCE, the sophists engaged in active debate and were both respected as popular intellectuals and ridiculed as dangerous teachers. After 399 BCE, however, Plato sharply dissociated Socrates from the sophists and treated his teacher as a model philosopher and the latter as fakes. While Plato's distinction was gradually accepted by his contemporaries and by subsequent thinkers through the fourth century BCE, some still regarded Socrates as a sophist, and many saw no clear distinction between philosophers and sophists. Following the discussion in the Academy, Aristotle finally defined the sophist as "one who makes money from apparent but not real wisdom" (Aristotle, *Sophistical Refutations* 1, 165a22–23). As this survey of the literature demonstrates, however, in antiquity the ambivalent view of Socrates prevailed.

## 3. Plato's Dissociation in the Fourth Century BCE: Rhetoric, Dialogue, Pedagogy

Among philosophers, Plato first made a sharp distinction between philosophers and sophists. Plato's innovation should be understood in relation to the controversy over Socrates's trial. We know that Polycrates published a pamphlet entitled *Accusation of Socrates* sometime around 393 BCE, in which the speaker Anytus claims that Socrates was guilty of political and educational crimes. Against this revived criticism, the pupils of Socrates, namely, Plato, Xenophon, Antisthenes, Euclides of Megara, Phaedo and Aeschines of Sphettus, wrote many Socratic dialogues. By depicting Socrates's conversations with people, the Socratic writers tried to demonstrate that he was a good person and thus show that his execution was unjust, as in, for example, the first book of Xenophon's *Memorabilia*.

This new movement in the Socratic literature in the first half of the fourth century BCE changed the literary context in which Socrates was discussed. The focus of these dialogues was on how to assess Socrates's life and activity, but at the same time his pupils competed and provided differing views of Socrates, often accompanied by tacit criticisms of other views. For example, we can see, in Xenophon's *Memorabilia* 2.1 and 3.8, that Socrates is censuring Aristippus for his way of life (Xenophon was more sympathetic to Antisthenes, who emphasized labour and autonomy). In addition, Plato's efforts to antagonize Antisthenes were well known in antiquity, though his name is mentioned only once in Plato's dialogues. In this way, each pupil of Socrates advocated his own position and original views on Socrates in rivalry with the others. The main strategy of Plato, for example, was to present Socrates as a philosopher by contrasting him with the other intellectuals, mainly the sophists. Unlike the other Socratics, this dissociation is clearest in the *Apology of Socrates*, but in several other dialogues Plato depicts Socrates arguing against Protagoras, Gorgias, Polus, Prodicus, Hippias, Thrasymachus, Euthydemus, Dionysodorus and other sophists. The dialogues of Plato thus created the traditional view that Socrates was a genuine philosopher whose views contrasted sharply with those of the sophists, who Plato depicts as pseudo-philosophers.

In contrast to Plato, other pupils scarcely concerned themselves with the problems of the sophists. They did not characterize Socrates in terms of philosophy, nor did they deny

that they themselves were sophists or under their influence. Indeed, we find that Plato was unique in identifying Socrates as a philosopher in contrast to sophists.[16] For example, Aristippus and Aeschines of Sphettus, two of Socrates's pupils, were sophists in that they were professional teachers of rhetoric who took fees for teaching. One anecdote shows that when Aristippus was condemned for taking money, he answered that Socrates did similar things. He likely meant that Socrates received financial support from his friends, including Crito.[17] Aeschines learned rhetoric from Gorgias and earned money from speechwriting and teaching rhetoric. It must have been natural for Aristippus, a visitor from Cyrene, and the poor Aeschines to earn money through their teaching activities, so both were called sophists by Aristotle, Lysias and others. Antisthenes was a pupil of both Socrates and Gorgias. Diogenes Laertius explains Antisthenes's career as if converted from Gorgias's rhetoric to Socrates's philosophy (see Diogenes Laertius, *Lives of the Eminent Philosophers* 6.1.1–2), but clearly this was not the case. Antisthenes was associated with Socrates from his youth, before Gorgias visited Athens and influenced the Athenians, and in his later writings he demonstrated the characteristics of both Gorgianic rhetoric and Socratic philosophy. In other words, for Antisthenes, being a sophist was not incompatible with being a pupil of Socrates. Likewise, Plato's contemporary teachers of rhetoric, namely, Alcidamas and Isocrates, never cared for this distinction and were proud of being both philosophers *and* sophists.

Xenophon was closer to Plato in that he criticized sophists in the *Cynegyticus*, perhaps with Aristippus in mind, but Plato was the only thinker who continued to criticize sophists. Plato tried to show the crucial difference between a sophist and a philosopher in his early and middle dialogues, but his final attempt in the *Sophist* deepened the examination. Not only does he try to give a definition of sophists in general, but he also casts fundamental doubt on the treatment of Socrates as a philosopher. The visitor from Elea and his interlocutor, Theaetetus, offer several definitions of sophist, but the sixth definition, called "noble sophistry" (*gennaia sophistikē*), reflects Socrates as this type of sophist, as being adept at purifying others' souls by refuting their beliefs and removing their stubborn self-conceit (Plato, *Sophist* 230A–231A).[18] Whether Socrates is indeed this type of sophist is hotly debated by commentators, but it is interesting and important to note that that Plato even raised the question about whether Socrates was a sophist again in this late dialogue. In addition, it should be kept in mind that Plato's criticism of the sophists was directed not only at sophists who were contemporaries of Socrates, such as Protagoras and Gorgias, but also at his own rivals, constituting two kinds, namely, his own sophistic contemporaries and other pupils of Socrates. After Plato, sophists of the second generation, such as Isocrates and Alcidamas, were active in the first half of the fourth century BCE, and many pupils of Socrates worked as sophists. Both kinds refused the Platonic distinction and instead believed that sophistry and philosophy were compatible pursuits. For these reasons, we cannot assume that Socrates was regarded as a philosopher, clearly different from sophists, even after his death.

## 4. Socrates in the Fifth Century BCE

By keeping our distance from Plato's distinction, which eventually became the traditional view of the history, we can see Socrates as one of the representative intellectuals (that is, the sophists) of the late fifth century BCE. By sharing activities and thought, rivalry and debate between Socrates and the sophists advanced philosophical thought in many respects. We can observe both common and distinct features between them and can consider some factors that led Plato to believe that Socrates was special.

To begin with, a common feature between Socrates and his contemporary thinkers, particularly the sophists, include, above all, the time and place in which they were active. In the latter half of the fifth century BCE, Athens was the centre of Greek culture, economy and politics. Plato had Hippias praise the city where he was staying as "the very shrine of wisdom in all Greece" (Plato, *Protagoras* 337D). Athenian democracy was established under Pericles's leadership, and intellectuals from different parts of Greece and from colonized

cities came to stay for various purposes. Thus Hippocrates of Chios, a leading mathematician, stayed in Athens while involved in litigation between 450–430 BCE. Herodotus is supposed to have been staying in Athens in this period, and Anaxagoras lived in Athens for twenty years (from 455–435 BCE) and was associated with Pericles. His pupil, the natural philosopher Archelaos, came to Athens and met the young Socrates in 452 BCE (Theodoret, *Greek Maladies* 12.66 = DK 60 A3 = LM ARCH. P6a). Democritus reported that he had visited Athens from Abdera and seen Socrates while nobody recognized this foreign philosopher (Diogenes Laertius, *Lives of the Eminent Philosophers* 9.7.38 = DK 68 A1 = LM ATOM. P21, P24). These examples indicate that Athens was the *polis* where scientists and philosophers gathered and exchanged ideas. Socrates, on the other hand, was a native thinker who never left the city except to fight in the Peloponnesian war.

Sophists were therefore central figures in the so-called golden age of Athens. Protagoras started the profession of the sophist around 455 BCE and often visited Athens to give lectures. Gorgias made his debut in the Athenian Assembly in 427 BCE, when he was sent as an ambassador by his city of Leontini. Thereafter, he exercised a great influence over Athenian people through his teaching and virtuosic displays of rhetorical skill. Other sophists, major and minor, stayed in this city to teach their pupils, of whom many were local citizens, but some came from different cities, as depicted in Plato's *Protagoras* (314E–315B). These sophists influenced many Athenian intellectuals, including Pericles, Euripides, Aristophanes, Thucydides, Critias and Alcibiades. From Plato's dialogues, particularly the *Protagoras* and *Gorgias*, we know the lively atmosphere of this period, so we can imagine the role Socrates played in his lifetime in Athens. He enjoyed an antagonistic dialogue with formidable sophists but also interacted and conversed with people from all areas of Athenian society. Foreign sophists and the Athenian Socrates therefore played an important role in the education of citizens in a democratic society.

In addition to sharing the Athenian *polis* as the locus of their activities, both sophists and Socrates engaged in speaking (*logos*); they spoke in front of large and small audiences and held dialogues (*dialegesthai*) in public or in private. We can better understand this feature of their activities if we compare it with the philosophical activities of the earlier natural philosophers, since the latter did not seem to have a common place to exchange their ideas in person. For example, we know no instance of Socrates conversing with or listening to Anaxagoras, who was staying in Athens when Socrates was young. It is likely that Anaxagoras did not give public speeches or engage in public discussion, though his thoughts were published in a book and widely read (Plato, *Apology of Socrates* 26D–E, *Phaedo* 97B–98B). By contrast, the sophists engaged in public speaking and professed to teach skills in public speech, and they prioritized speaking over writing. Although we know that Protagoras wrote the treatises *Truth* and *On Gods*, in addition to a dozen works whose titles are reported in Diogenes Laertius (*Lives of the Eminent Philosophers* 9.8.55 = DK 80 A1 = LM PROT. D1), we have only scant evidence of writings by sophists, probably because they emphasized *ex tempore* speech and were thus less interested in writing scholarly treatises on theoretical themes.[19] They produced writings only for specific purposes. For example, two epideictic works by Gorgias, *Encomium of Helen* and *Defense of Palamedes*, aim to demonstrate his skills in rhetoric, likely to recruit pupils. Prodicus's *Choice of Heracles* and Hippias's *Trojan Dialogue* were also texts for recital. The most notable example is Antiphon's *Tetralogies*, which were likely composed to display a variety of arguments for rhetoric education. In sum, their writings were subsidiary teaching materials rather than main achievements of their profession and inquiry. In light of this oral tradition, we can see that Socrates was not unique in having produced no written texts. He engaged in dialogue and discussed various topics with his fellow citizens and foreign guests, as sophists did, but the obsession with writing became stronger only in the fourth century BCE, when Isocrates, Alcidamas and Plato argued over the role of writing and competed with each other in writing.[20]

Yet another shared interest of Socrates and the sophists were the arts of dialogue and refutation. Socrates engaged in dialogue, which is often contrasted with the long

speeches given by sophists. It is true that while Protagoras, Gorgias, Antiphon and others focused on forensic speeches, not all sophists were interested in this genre all the time. Hippias and Prodicus, for example, seemed to be better at political speeches. In this respect, however, Socrates was uninterested in speaking in court or the assembly, though he seems to have given a wonderful defence at his trial. Socrates developed his speaking skills in the form of dialogue, but this, too, was also part of the sophists' repertoire. Some sophists, such as Euthydemus and Dionysodorus in Plato's *Euthydemus*, were skilled at short questions and answers. Instead of giving a lengthy, one-sided speech, the cross-examination through questioning—a particular form of dialogue—was developed into the method of "refutation" (*elenchos*). We know from Aristotle's *Sophistical Refutations* that refutations, used by sophists, often entail the use of fallacies. In this respect, both Socrates and sophists shared the same form of argumentation. In *Clouds,* Aristophanes parodies the sophistic and Socratic practice of staging competing arguments (*dissoi logoi*) by presenting a contest between the anthropomorphic Stronger Argument and the Weaker Argument (which he also calls the "unjust", "wrong", and "non-paying" argument). Plato formalized the art of dialectic (*dialektikē*) from the dialogue and refutation practiced by Socrates, but insofar as dialectical arguments are practiced by two parties—the questioner and answerer—to attack and defend the same thesis, it was clearly developed from Protagoras's art of antilogy (*antilogikē*) or "discussing both sides" of an issue, which itself originated from Zeno's Eleatic argumentation (see Reames and Giombini in this issue). Considering this wider context, we should view the Socratic *elenchus* and the Platonic dialectic not as original inventions but as extensions of the sophistic arts of speech and argumentation.

Here, it is important that both sophists and Socrates focused on speech (*logos*) and discussed its art. As Edward Schiappa demonstrates ([Schiappa 1991](#), ch. 3), the technical term *rhētorikē* was probably not used by the sophists in the fifth century BCE, and they instead seem to have called their own skill *technē logōn* (art of speeches). While the extant fragments of the sophists do not mention this phrase, Plato and Xenophon often used it to describe the sophists' skills. Xenophon, for example, introduced an episode of the conflict between Socrates and the Thirty in which Charicles and Critias forbid the "teaching of *technē logōn*" (Xenophon, *Memorabilia* 1.2.31, 34), a barb aimed at Socrates. Plato also used the phrase to describe the various skills of the sophists and rhetoricians in the *Phaedrus*. The designation *technē logōn* is ambiguous, embracing logical arguments and rhetorical speech, a skill in which both Socrates and other sophists were experts. This is why Socrates was described as a formidable speaker (*deinos legein*), along with the other sophists, and why the Thirty handed down the ban on Socrates. Later, Plato and Aristotle made a clear distinction between rhetoric and logic and between fallacies and valid arguments, but before this, sophists and Socrates were all regarded as experts in speech, i.e., practitioners and teachers of *technē logōn*. Socrates discussed any topic without preparation with people he met, much like the *ex tempore* speech with which Gorgias entertained Athenian crowds. Time, place and occasion (*kairos*) are the most important elements of sophists' speaking skills, but Socrates also seems to have been a master of this important skill. In this way, Socrates' dialogue had much in common with the art of speech that sophists claimed to possess and teach.

Finally, another important common interest shared by Socrates, Protagoras and other sophists (but not Gorgias) was the teaching of virtue or excellence (*aretē*). Socrates likely never claimed to *teach*; nevertheless, he always encouraged others, through dialogue, to care for the soul and for virtue (Plato, *Apology of Socrates* 31B, 41E). Thus, a central concern of both the sophists and Socrates was virtue or excellence. Now, we must recognize that Socrates was not the first philosopher to discuss moral issues, as Cicero claimed, because sophists and Socrates formed a shared new interest in ethics in this period. Socrates encouraged his interlocutors to care for the soul (*psychē*), and although the meaning of *psychē* had changed a great deal since Homer's epic, Socrates was most responsible for the change to its meaning. Nevertheless, in his praise of the power of *logos* in the *Encomium of Helen,* Gorgias also discusses the "soul" as the object of influence or manipulation.

Besides, Plato characterizes rhetoric as the "leading of souls" (*psychagogein*) in the *Phaedrus* (261A, 271C), but we should note that this mysterious word was also used satirically to describe Socrates in Aristophanes's *Birds*. The chorus sings about Socrates evoking the souls (*psychagogein*), together with his pupil Chaerephon: "Near by the land of the Sciapodes there is a marsh, from the borders whereof the unwashed Socrates evokes the souls of men./Pisander came one day to see his soul, which he had left there when still alive./He offered a little victim, a camel, slit his throat and, following the example of Odysseus, stepped one pace backwards./Then that bat of a Chaerephon came up from hell to drink the camel's blood" (1553–1564). Moreover, the emphasis on love (*erōs*) is usually believed to characterize Socrates's philosophy. In particular, his pupil Aeschines highlighted this Socratic theme in his dialogues, but the sophists were not unfamiliar with this theme either. Gorgias also discusses the power of *erōs* as the fourth factor that may have coerced Helen to fly to Troy, and he uses the power of love as analogy for the power of his own speech as an irresistible force and charm.

## 5. Provocative New Ideas

In addition to a sharing an interest in the arts of *logos* (rhetoric, dialogue, refutation, and argument) and the teaching of *aretê*, the sophists and Socrates—unlike the natural philosophers—shared a concern for society, language and morality. While the conversion of nature and gods to humans is a hallmark of this era, Protagoras's famous thesis that "man is the measure of all things" (DK 80 B1) may sound similar to the "human wisdom" Plato attributes to Socrates in the *Apology*. In this sense, again, Socrates was not the first thinker who considered human issues. Sophists advocated many new ideas, including many seemingly paradoxical theses. Gorgias was especially famous for *paradoxologia*, as we can see in his extant works. Notable in this regard is his claim in *On Nature: Or, On Not-Being* (and I paraphrase) that "nothing exists; even if it does, it cannot be known; even if it can be known, it cannot be communicated to others". In this respect, however, Socrates also presented many apparently paradoxical statements, such as "no one does wrong willingly", "being harmed is better than doing harm", "virtues are one", and "virtue is knowledge", which did not stand alone but functioned as counterarguments to our common sense. Provocative and shocking arguments were an effective means of astonishing, inspiring and fascinating audiences, and the sophists and Socrates alike entertained similar arguments and aimed for comparable effects. In fact, new ideas advanced by Socrates, including a critical examination of democracy and traditional religion, were noted as possible causes of corrupting youth and damaging traditional morality. The more popular and influential they were, the more dangerous people believed them to be. Indeed, Socrates seems to have been viewed as more provocative than any other sophist in cross-examining traditional values and morals and in this way committing impiety. Although there is a contrast between the polymathy or omniscience declared by the sophists and the ignorance or disavowal of knowledge asserted by Socrates, few Athenians may have taken them at face value. After all, Socrates's famous confession of ignorance was often considered an ironic mask for concealing his wisdom, or even as another paradox.

Despite all these common features and shared interests, Plato began dissociating Socrates from the other sophists. Here, the new concepts of the philosopher (*philosophos*) and philosophy (*philosophia*), which had a Pythagorean origin, came to play a crucial role. Plato saw engaging in dialogue as one of the criteria defining a true philosopher: caring for the soul and preparing for death now constituted the essential task (and image) of the philosopher. Several other features, notably confronting *aporia*, confessing ignorance, using irony, practicing the maieutic method, not taking fees, and caring for fellow citizens also characterize Socrates as a philosopher in the dialogues of Plato. However, Plato's efforts to characterize Socrates led to the theory of Forms, through which he defines (in the *Republic*) the philosopher as one who loves to see the truth, namely, the transcendent Forms. Socrates's role was thus changed to that of a philosopher, a seeker of wisdom who left the cave, caught a glimpse of real Being, and then returned to the cave at the risk of being

killed by ignorant people who misunderstood him. From the traditional view established by Plato, the differences between the philosopher Socrates and the sophists appear obvious, but we must again ask on what basis Socrates and the sophists can be separated.

## 6. Antiphon and Socrates

To conclude, I would like to focus on one particular sophist and his relation to Socrates, namely, Antiphon of Rhamnus. Although there is controversy surrounding the identity and plurality of Antiphon(s), the unitarian view recently seems stronger than the separatist view.[21] On the unitarian side, I believe we can observe Antiphon as one of the most important thinkers of the fifth century BCE. If this is correct, then, in addition to being a teacher of rhetoric and an oligarchic statesman, he was the author of the treatise entitled *On Truth* and other books and the sophist whom Xenophon confronted with Socrates in *Memorabilia* 1.6. Antiphon's rhetorical works, *Tetralogies*, are epideictic pieces that make full use of *eikos* arguments (arguments from likelihood or plausibility) which are supposed to have been introduced by Tisias in Sicily a few decades earlier.

The most interesting fact about Antiphon of Rhamnus is that he was an Athenian citizen, a senior of Socrates and a rhetorician and sophist of Athenian origin. Plato never mentions his name except in one passing reference to him as an inferior teacher of rhetoric in the *Menexenus*, the political and rhetorical dialogue that presents the famous funeral speech of Athens. Socrates speaks about his teachers: "These [Aspasia and Connus] are my two instructors, the one in music, the other in rhetoric. So it is not surprising that a man who is trained like me should be clever at speaking. But even a man less well taught than I, who had learnt his music from Lamprus and his rhetoric from Antiphon the Rhamnusian,—even such a one, I say, could none the less win credit by praising Athenians before an Athenian audience" (*Menexenus* 235E–236A; trans. W. R. M. Lamb). It is interesting that Socrates contrasts Aspasia, the partner and advisor of Pericles, with Antiphon. Some commentators suspect that Plato had Thucydides in mind here, because Antiphon is sometimes said to be the teacher of this Athenian historian. I suspect the clear neglect of Antiphon in his entire dialogues to be an interesting sign of Plato's difficulty in discussing this fellow citizen. Plato defines the sophists as travelling teachers responsible for their harmful influence on Athens, which stands in sharp contrast to his portrayal of Socrates as a resident patriot responsible for genuine education. In this sense, Plato's characterization of the sophists fails to accurately portray Antiphon. Under Plato's influence, we may believe that Socrates was the first native philosopher of Athens, but if Antiphon is included among the fifth-century sophists, he preceded Socrates as a provocative thinker born and active in Athens.

In addition, Antiphon's anti-democratic position may appear similar to that of Socrates. Antiphon was tried after the failure of the oligarchic revolution of the Four Hundred in 411 BCE. He did not escape Athens but instead made a brilliant defence in court, only to be sentenced to death. The charge was purely political, but we can imagine that for contemporary Athenians the trial and execution of Socrates were reminiscent of those of Antiphon twelve years earlier. While the trial of Antiphon was a nightmare of the abortive attempt to overturn Athenian democracy, the trial of Socrates became another nightmare of the Thirty tyrants in 404–403 BCE and marked a failure of the restored democracy. Thucydides praised Antiphon's defence speech, *On Revolution* (Genève papyrus, *CPF* 264(2)–267; Thucydides, *Histories* 8.68.2 = LM ANTIPH. P15), and Socrates's speech became a masterpiece in the form of Plato's *Apology* (though it is uncertain how much came from the historical speech). Their political charges were perhaps intertwined with the charge of being sophists. But the main reason we completely ignore Antiphon, forerunner of Socrates, is that Plato erased him from the history of philosophy and of the sophists. Nevertheless, we see in Xenophon's *Memorabilia* three short conversations between the sophist Antiphon and Socrates. The first scene shows that Antiphon challenged Socrates as a rival of his profession and contrasted his own sophistic teaching with the "philosophy" of Socrates. Antiphon, if he was the Athenian sophist, was thus a shadowy double of Socrates in democratic Athens.

### 7. Conclusions

We are facing a mystery concerning the historical gap between the fifth and fourth centuries BCE concerning Greek views on the difference, if any, between Socrates and the sophists. We modern readers, under the influence of the Platonic dialogues, tend to see a clear distinction between Socrates the philosopher and the sophists. As I have shown, however, the history is far more complicated. On the one hand, it is anachronistic and misleading to assume that this distinction was prevalent in the fifth century BCE. Socrates obviously shared an interest in and skills of the intellectual activities with the sophists, and he himself was counted among the sophists by his fellow Athenians. On the other hand, the crucial difference that Plato detected and continued to demonstrate between Socrates and the sophists became the traditional view in the history of rhetoric and philosophy—and modern readers firmly support it. But if we are to undertake the important task of reconsidering the sophists, we should return to the threshold of the fifth and fourth centuries BCE, when Socrates was executed as a sophist. His death should remind us that questions about sophists are in fact questions about philosophy.[22]

**Funding:** This research received no external funding.

**Institutional Review Board Statement:** Not applicable.

**Informed Consent Statement:** Not applicable.

**Data Availability Statement:** Not applicable.

**Conflicts of Interest:** The author declares no conflict of interest.

## Notes

1.     For the label and meaning of "sophists" see Notomi (2012).
2.     Guthrie explains the reason that "the Press hopes to make them more easily and cheaply available to students" (Guthrie 1971a, p. 1, 1971b, p. 1).
3.     For the novelty, see Rossetti (2016, pp. 162–63).
4.     André Laks informs me that they did not have the clear intention of changing the traditional treatment. He added that the volume title, *Sophists*, is an editorial convenience.
5.     Laks (2018), different from his edition, defends this traditional title. Interestingly, Friedrich Nietzsche uses the term "The Pre-Platonic Philosophers" (*Die Vorplatonische Philosophen*).
6.     The omission of Critias is appropriate because he was not a sophist by any definition, but he was later included in the list after Philostratus's *Lives of the Sophists*; cf. Notomi (2000). Laks and Most put the Sisyphus fragment (Critias?) in the chapter, *Dramatic Appendix* (T63), but two minor sophists, Lycophron (contemporary with Plato) and Xeniades (date uncertain), are misplaced, and the order would become more consistent if we exclude them from among the fifth-century sophists.
7.     LM SOC. P13–P14 and D62–D65 briefly mention his pupils.
8.     I extensively discuss this idea in Notomi (2017).
9.     The name Polycrates is not mentioned in Xenophon, but Carel Gabriel Cobet's suggestion that the "accuser" (single) is not Meletus in the actual trial but the Anytus of Polycrates's pamphlet is now widely accepted. See Dorion and Bandini (2000, pp. 79–81).
10.    Cf. Dorion and Bandini (2000, p. 60) and Macleod (2008, pp. 127–28). Xenophon used the word "sophist" only in the following places: *Memorabilia* 1.1.11, 1.6.1, 1.6.13, 4.2.1; *Symposium* 4.5.1; *Cynegeticus* 13.1.1, 13.1.4, 13.6.2, (13.7.2 adjective), 13.8.1, 13.8.4, 13.9.1, 13.9.2; *Cyropaedia* 3.1.14, 6.1.41; *Ways and Means* 5.4.2.
11.    Aristotle, *Rhetoric* 1402 a 24–28 (DK 80 A 21). See also Eudoxus, fr. 307 Lasserre; Steph. Byz. *Ethnica*, A6, 1.18.13–5 Billerbeck (DK 80 A 21). For the complex relationship between Socrates and Protagoras, see Corradi (2017).
12.    See Aristophanes, *Clouds* 331 and 1111, where Socrates promises to turn Pheidippides into a clever sophist, and 1309.
13.    In the *Symposium*, Plato depicts the scene in which Socrates and Aristophanes had a friendly chat.
14.    Indirect evidence is seen in Libanius, *Defence of Socrates*, 16, 38, 52, 15–159.
15.    The main testimonies of Aristophanes, Ameipsias, Callias, Eupolis and Telecleides are collected in LM DRAM. T26–T32.
16.    Notomi (2010) discusses the following cases.
17.    Diogenes Laertius, *Lives of the Eminent Philosophers* 2.7.65, 74.
18.    For the problem of "sophist of noble lineage", see Notomi (1999, pp. 64–68, 275–77).

[19] O'Sullivan (1996) discusses the important role of writing in the sophists.

[20] See the intertextual relations between Plato's *Phaedrus*, Alcidamas's *On the Sophists* and Isocrates's *Against the Sophists*.

[21] A single Antiphon is proposed by Michael Gagarin, Decleva Caizzi and Laks and Most, whereas Gerard Pendrick strongly defends the plurality of Antiphons. However, the arguments against the single Antiphon in Pendrick (2002, pp. 1–26) are not convincing.

[22] The main part of this essay was presented at the Virtual Socrates Colloquium of the International Society for Socratic Studies on 2 September 2020. I thank the organizer, Donald Morrison, and the participants, especially Andreé Laks and Debra Nails, for valuable comments.

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
