# Peer review of "Socrates and the Sophists: Reconsidering the History of Criticisms of the Sophists"

_humanities, doi:10.3390/h11060153_

Round 1
Reviewer 1 Report
This paper asks what a sophist is and who Socrates is, and succeeds in illustrating how we can approach these important questions by combining them harmoniously. The reader can learn a lot from the author’s historical approach with a huge amount of knowledge of ancient writers, e.g. Aristophanes and Xenophon. For example, the author’s suggestion sounds convincing in arguing that “a major division should be made between the natural philosophers in Ionia and Italy and the sophists and Socrates, active in Athens” (128-129). Among other things, personally, I found the author’s remark on the word “psychagogein” in Aristophanes’ Birds (478-486) very interesting.
Besides, the overall structure of the paper is illuminating, first, by contrasting Plato’s view on Socrates with those of other pupils of Socrates in the fourth century BCE, and, secondly, by clarifying various similarities between Socrates and sophists in the fifth century BCE. This double structure leads us to be free from Plato’s strong influence on our view of the historical Socrates. To reconsider the relation between Socrates and the sophists, as the author concludes, “we should return to the threshold of the fifth and fourth centuries” (600-601). In this sense, Section 6 may be the most original of the author’s points that contribute to our understanding of Plato’s depiction of Socrates. For this paper persuasively elucidates why Plato almost ignores Antiphon on purpose in his works. This claim seems to be based on the unitarian view, so the author might add a couple of sentences to argue even briefly against Pendrick’s separatist view in footnote xx.
Author Response
Thank you very much for the valuable comments. In response to your suggestion concerning the identity of Antiphon(s), I added a sentence in the final footnote, although I cannot give a full argument there.
Reviewer 2 Report
A good paper, with a clear structure and sound argument, arguing for an interpretation of Socrates as a sophist. The idea is not new, but it is gaining traction after Laks/Most edition. The paper collect the most relevant evidence and make a very compelling case. Even though it is not decisive, more attention could be given to Plato. The paper clearly rejects Plato's view, but do not properly discuss it: why and how Plato presented Socrates as different from the Sophist? By reading the paper this is not clear – it seems simply impossible that somebody could even think of Socrates as not being a sophist. A quick discussion of Plato might help better understand what was at stake.
Two minor points:
at line 367: I doubt that what the Platonic character Hippias says in the Protagoras can be taken as an uncontroversial evidence for Hippias.
at line 578: I doubt that Plato's apology does mirror what Socrates precisely said. How to account, then, for Xenophon's apology, which is different? That Socrates gave a very poor speech according to Plato is confirmed also by a famous anecdote (for what it is worth). On this point it is hard to compare Socrates and Antipho.
Bibliography is not always updated
Author Response
Thank you very much for the valuable comments. I have made the two changes you pointed out. Please see lines 368 and 581. Regarding your main point, I'd like to emphasize that Plato was the only one among Socrates' pupils who strictly contrasted philosopher and sophist. Finally, I have changed the bibliography.
Reviewer 3 Report
The paper provides an interesting and original analysis of the relation between Socrates and the sophists. The author argues persuasively that Socrates belonged to the sophistic movement before Plato separated him form the other sophists. In this context the relation of Socrates to Antiphon is carefully studied. The matter discussed in the paper is original, well-argued, and convincing. The paper is clearly written. I just have some minor observations:
- at p. 1, a brief discussion of the different meanings of the term “sophist” and their evolution would be probably necessary: see now G. Ramírez Vidal, La invención de los sofistas (Universidad Nacional Autónoma de México: México, 2016).
- at p. 4, on the relationship between Protagoras and Socrates M. Corradi, Protagorean Socrates, Socratic Protagoras: a Narrative Strategy from Aristophanes to Plato, in A. Stavru, C. Moore (eds.), Socrates and the Socratic Dialogue (Leiden-New York: Brill, 2017), pp. 84-104, and L. A. Dorion, ‘De quelques positions communes aux sophistes et au Socrate de Xénophon’, Philosophia, 50, 2020, pp. 79-94, should be considered.
- at p. 8, it is probably true that the sophists “emphasized ex tempore speech and were thus less interested in writing scholarly treatises on theoretical themes”, but the importance the sophists attached to writing should not be underestimated: see N. O’Sullivan, ‘Written and Spoken in the first Sophistic,’ in I. Worthington (ed.), Voice into text. Orality and Literacy in Ancient Greece (Leiden: Brill, 1996), pp. 115-127, and M. S. Harbsmeier, ‘Die Überlieferung der sophistischen Literatur in der Antike’, Proceedings of the Danish Institute at Athens, VI, 2009, pp. 285-308.
Once this minimal revision has been made, I strongly recommend the publication of the article.
Author Response
Thank you very much for the valuable comments on the additional references. I have added some references to the footnotes and bibliography, although I cannot include them all. Some works are hard to obtain. I will definitely study them soon. Thank you once again.